# Fossil or Non-Fossil: A Case Study in the Archaeological Wheat *Triticum parvicoccum* (Poaceae: Triticeae)

**DOI:** 10.3390/genes16030274

**Published:** 2025-02-25

**Authors:** Diego Rivera, P. Pablo Ferrer-Gallego, Concepción Obón, Francisco Alcaraz, Emilio Laguna, Nikolay P. Goncharov, Mordechai Kislev

**Affiliations:** 1Departamento Biología Vegetal, Facultad Biología, Universidad de Murcia, 30100 Murcia, Spain; falcaraz@um.es; 2Servicio de Vida Silvestre y Red Natura 2000, Centro para la Investigación y Experimentación Forestal (CIEF), Generalitat Valenciana, Avda. Comarques del País Valencià 114, Quart de Poblet, 46930 Valencia, Spain; flora.cief@gva.es (P.P.F.-G.); laguna_emi@gva.es (E.L.); 3CIAGRO, Escuela Politécnica Superior de Orihuela, Universidad Miguel Hernández, Ctra. Beniel, Km 3,2, 03312 Orihuela, Spain; cobon@umh.es; 4Institute of Cytology and Genetics, Siberian Branch of Russian Academy of Sciences, Novosibirsk 630090, Russia; gonch@bionet.nsc.ru; 5The Mina and Everard Goodman Faculty of Life Sciences, Bar-Ilan University, Ramat-Gan 5290002, Israel; mordechai.kislev@biu.ac.il

**Keywords:** archaeobotany, fossil, holotype, nomenclature, non-fossil, palaeoethnobotany, taxonomy, cultivated wheat origins

## Abstract

Background/Objectives: The archaeobotanical taxon “*Triticum parvicoccum*” was first described in 1980 as a small-grained, naked, free-threshing, and dense ear tetraploid wheat species (2*n* = 4*x* = 28) identified from archaeological remains. This primitive tetraploid, cultivated in the Levant approximately 9000 years ago and subsequently dispersed throughout the Fertile Crescent, represents a potential contributor of the BBAA genomes to *T. aestivum*. This study aims to resolve the complex nomenclatural status of this taxon, which has remained ambiguous due to competing interpretations under fossil and non-fossil taxonomic regulations. Methods: We conducted a comprehensive nomenclatural review to evaluate the taxonomic validity of *T. parvicoccum*, analyzing previous research on the classification of archaeobotanical materials in relation to fossil status. Results: Our analysis demonstrated that archaeobotanical materials do not qualify as fossils and led to the validation of the taxon at a subspecific rank as a non-fossil entity: *T. turgidum* subsp. *parvicoccum* Kislev. subsp. nov. The holotype was established using a charred rachis fragment from Timnah (Tel Batash), an archaeological site on the inner Coastal Plain (Shfela) adjacent to the western piedmont of the Judean Mountains, Israel. Conclusions: This study resolves the longstanding nomenclatural uncertainty surrounding this archaeologically significant wheat taxon, providing a valid taxonomic designation that reflects its biological and historical importance while adhering to current botanical nomenclature standards.

## 1. Introduction

Since the beginning of the 19th century, archaeobotanical studies have been devoted to the identification of plant remains, especially the fruits and seeds of cultivated plants. As a result, thousands of archaeobotanical papers have been published [1,2,3,4]. Archaeobotanical studies have focused on the identification of plant remains in the sense of current species and only occasionally have new taxa been published based on archaeological material, with one of the most notable being “*Triticum parvicoccum* Kislev” [5,6,7].

Archaeobotany occupies a unique interdisciplinary position at the intersection of paleontology, botany, and weed science. This complex positioning necessitates careful consideration of taxonomic nomenclature to prevent the emergence of “zombie-taxa”—entities that oscillate ambiguously between fossil and non-fossil classifications. All taxonomic designations based on archaeobotanical material must adhere to the International Code of Nomenclature for algae, fungi, and plants (ICN or Shenzhen Code) [8], regardless of their classification status.

The Shenzhen Code establishes clear guidelines through Art. 13.3, stipulating that nomenclature defaults to non-fossil classification unless the type specimen exhibits fossil origins as defined by Art. 1.2. The determination of fossil status primarily relies on stratigraphic relationships at the site of original discovery, with provisions for non-fossil taxa applying in cases of ambiguous stratigraphic context, as well as for all diatoms.

Archaeobotanical specimens primarily present in three preservation states: desiccated, waterlogged, and carbonized. The classification of these materials as fossil taxa requires both a fossil-type specimen (Art. 1.2) and clear stratigraphic relationships within the discovery site (Art. 13.3)—specifically, originating within a distinct stratigraphic rock layer. Consequently, carbonized plant materials only qualify as fossil taxa when their type specimens originated within defined stratigraphic rock formations. This distinction becomes particularly relevant for chronological classification: specimens from ancient geological periods (e.g., Pliocene, Miocene) may qualify as fossils, while Holocene specimens are considered recent, non-fossil materials. In palaeobotanical contexts, these more recent specimens are often designated as “subfossils” and are taxonomically subordinate to contemporary non-fossil taxa [9].

The taxonomic classification of small-grained naked wheat specimens from archaeological contexts presents a particular challenge for both taxonomists and archaeobotanists, as these morphotypes lacked contemporary analogs in living populations or herbarium collections until recent discoveries. This absence of modern reference material led to the establishment of new taxa based on archaeobotanical likely extinct-type specimens [10,11].

*Triticum* L. (Poaceae: Triticeae Dumort.) is a very complex genus of approximately 25 wild and domesticated species [12]. Currently, the classification of wheat species is artificial and very subjective, even with the molecular genetic tools currently available [13]. One possible cause of this low taxonomic resolution is the relative evolutionary youth of cultivated wheat species, all of which emerged in the last 6000–12,000 years [14].

The infrageneric taxonomy of *Triticum* has been published by several authors [15,16,17,18,19,20,21,22,23,24]. Currently, the genus is divided based on ploidy levels, cytoplasm types, and genome structure and composition [24,25,26,27,28,29,30,31,32]. Molecular genetics analyses have supported the sectional divisions [29]: sect. *Monococcon* Dum., *Dicoccoides* Flaksb., *Triticum*, *Timopheevii* A.Filat. et Dorof., and *Compositum* N.P. Gontsch [23,24]. However, other taxonomists [33,34,35] have recommended placing the di- (2*n* = 2*x* = 14), tetra- (2*n* = 4*x* = 28), and hexaploid (2*n* = 6*x* = 42) wheat species in three different genera.

Zohary in 1969 [36] observed that it was not until wheat cultivation reached Armenia, Transcaucasia, and the Southern Caspian region that plants of cultivated tetraploid wheat came into contact with diploid *Aegilops tauschii* Coss., the D-subgenome donor of hexaploid wheats. Since the process of wheat hexaploidization took place in the ancient past, scientists referred to short round-grained archaeological materials as bread wheat and, consequently, they determined that they are hexaploid. For example, Heer [10,11] referred to wheat samples of the Swiss Neolithic as *T. vulgare* var. *antiquorum* Heer. Flaksberger [37] suggested that this variety should be transferred to *T. compactum* Host and named it *T. compactum* var. *antiquorum*. Kislev [38] discussed the findings of *T. vulgare* [var.] *antiquorum* in Central Europe and ascribed them to the hexaploid wheats group because of their morphological similarities. Further, he proposed an explanation for their origin as the result of the amphidiploidization of a tetraploid *T. parvicoccum* and diploid *A. tauschii* hybrid.

We focus here on resolving the nomenclatural challenges surrounding *Triticum parvicoccum*, a taxon whose unique circumstance of being described solely from archaeobotanical specimens has complicated its taxonomic classification. As it is based on carbonized archeological material alone, no equivalent genetic information is available [39].

## 2. Materials and Methods

Nomenclatural Analysis: We conducted a comprehensive review of the taxonomic literature pertaining to “*Triticum parvicoccum* Kislev”, including the original description by Kislev in 1980 [38] and subsequent treatments. The analysis focused on evaluating the taxon’s status under both the fossil (Art. 13.3 of the ICN) and non-fossil provisions of the International Code of Nomenclature for algae, fungi, and plants (ICN or Shenzhen Code) [8]. We examined all relevant nomenclatural acts and publications from 1980 to present to trace the taxonomic history of the taxon.

Nomenclatural Procedures: The formal nomenclatural validation procedure followed Articles 32–45 of the International Code of Nomenclature [8], with particular attention to the requirements for valid publication of names of non-fossil taxa. We established the nomenclatural status through careful examination of:The original publication’s compliance with Art. 32.1 of the ICN for valid publicationThe legitimacy of the name under Art. 6.5The correct form of the name according to Art. 21 and 24 for subspecific ranksThe requirements for type designation under Art. 40

Archaeological Material: The holotype specimen consists of a charred rachis fragment recovered from archaeological excavations at Timnah (Tel Batash) (31°47’ N, 34°54’ E), located in the inner Coastal Plain of Israel (Figure 1). The specimen was retrieved from secure archaeological contexts dated to approximately 3150 ± 20 BP. This material was recovered in the season of excavation conducted during the year 1979 by a consortium of U.S. institutions of higher learning in collaboration with The Hebrew University of Jerusalem, notably in basket 4120 for the type, but also 4132, 4142, and 4143.

The original material was initially examined using both the Olympus SZX10 stereomicroscopic system (Olympus Corporation, Tokyo, Japan) and scanning electron microscopic techniques Philips SEM 505 (Philips High Tech Campus, Eindhoven, The Netherlands) available at Bar-Ilan University (Israel) and Weizmann Institute to verify diagnostic morphological characteristics, as described in the protologue.

Repository Information: The holotype specimen is permanently housed in the Archaeological Collections of Archaeobotany Laboratory, National Natural History Collection of Seeds and Fruits at Bar-Ilan University (Israel). Digital microscopic images and detailed documentation are available through the institution’s webpage https://cris.biu.ac.il/en/equipments/national-natural-history-collection-of-seeds-and-fruits-at-bar-il (accessed on 16 February 2025).

Taxonomic Assessment: In different stages from 2021 to 2024, we evaluated the specimen’s preservation state and taphonomic conditions to determine its status relative to the concept of fossils as defined by the Code. This included comparative analysis with other archaeobotanical wheat remains from contemporary sites in the Fertile Crescent region. The morphological characteristics were assessed against the diagnostic features of known tetraploid wheat taxa (*Triticum turgidum sensu lato*) to confirm its taxonomic placement.

## 3. Results

### 3.1. The Oldest Free-Threshing Tetraploid Wheat

When considering the small-grained (length < 5 mm) naked wheat finds from West Asia, as well as their antiquity (preceding the emergence of hexaploid wheats), Kislev [38,40] proposed that these samples were tetraploid. They were characterized by their small grains, dense ears, and short, narrow internodes, and named them “*T. parvicoccum*”, properties that are characteristic of Mediterranean climate areas [38,41,42]. Regarding the studies of Martínez-Moreno et al. [43], it is possible that the early free-threshing wheats where not durum wheat (*T. durum* Desf.), but the extinct tetraploid (mentioned as “*T. turgidum* subsp. *parvicoccum*”) that was grown until the first century CE.

This taxon is only known from archaeological materials [6,44,45]. Thus, the IFPNI-International-Editorial-Board [5] reports it as fossil, and as “Repository Bar-llan University, Department of Life Sciences, Ramat Gan, Israel”, with repository number (or code): Tel Aphek, locus 1700. For GBIF [46], the taxon is a “doubtful species”; for the ArchbotLit [7] database, “*T. parvicoccum*” is a “valid name”; and for The Plant List [47], it is an “unresolved name”, and for POWO [48], IPNI [49] and GRIN Taxonomy [50] it merely does not exist. IPK [51], however, accepts the name as a synonym of *T. turgidum* subsp. *durum* (Desf.) Husn. The EuroMed Plant Base [52] accepts *T. parvicoccum* as a non-fossil, presently extinct, species endemic to the Easten Mediterranean, Balkans, and West Asia. According to Feldman and Levy in 2023 [53] (p. 388) *T. parvicoccum* is a “domesticated fossil taxon” and alternatively (pp. 402–405) it is a subspecies of the living taxon *T. turgidum* and considered as an extinct archaeobotanical, free-threshing, tetraploid wheat taxon, and Hammer and Knüpfer [54] and Hammer and Khoshbakht [55] abound in the idea of being *T. parvicoccum*, an extinct wheat species.

“*Triticum parvicoccum*” was described initially by Kislev in 1980 (p. 97) [38] as an “archaeobotanical species”, but he gave no Latin diagnosis. For his description of the new taxon, Kislev [38] integrated grains and rachis fragments found in proximity (e.g., in Tel Aphek), presumably belonging to the same specimen. This author indicated four internode images of material recovered from Tel Aphek, locus 1700 as the “holotype”. The taxonomic status of this name presents a complex nomenclatural challenge under the International Code of Nomenclature [8] (ICN). Since the name lacks a Latin description or diagnosis, its validity hinges on whether “*Triticum parvicoccum*” is classified as a fossil or non-fossil taxon. Art. 43.1 of the ICN invalidates the publication if the species is considered non-fossil, despite it being extinct and known exclusively from archaeobotanical remains. Interestingly, classification as a fossil taxon would circumvent this restriction, as the ICN applies less stringent requirements to fossil taxa.

The publication requirements for fossil plant taxa in 1980 were governed by the then-current 1978 edition of the International Code of Botanical Nomenclature (ICBN) [56]. These regulations differed from those applying to non-fossil taxa, particularly regarding descriptive and illustrative requirements. Notably, the Latin diagnosis requirement could be waived if the fossil taxon name was accompanied by sufficient illustrations and explanatory material that distinguished it from related taxa. Furthermore, Art. 59’s special provisions for fossil plants acknowledged the frequently fragmentary nature of fossil material by implementing more flexible rules compared to those for extant taxa.

We must also consider the well-preserved spike remnants found later, which were not included by Kislev [38] in his first diagnosis. These contributed to a clearer definition of the taxon [57,58]. The finds include charred free threshed wheat from Timnah (Tel Batash), dating to the Late Bronze Age IIA. To straighten out the different approaches to naming this archaeobotanical wheat, we provide the necessary requirements to validate its name at subspecific rank, as “*Triticum turgidum* subsp. *parvicoccum”* (see below) [Kislev [57] proceeded in 2009 to catalog it as a subspecies of *T. turgidum* L., following the Anglo-Saxon scheme for *Triticum* taxonomy].

According to Kislev [38,42], the chief rationale for allotting “*T. parvicoccum*” the rank of species rather than a variety, is that no known tetraploid wheat has such small grains. However, according to Bowden [15] (p. 669), this species should be included in *T. turgidum* L. along with the rest of tetraploid wheats: see Vavilov [59] for phytogeographic arguments supporting Bowden.

More recently, it was also suggested that “*T. parvicoccum*” did not deserve the rank of a species, but only a form or subspecies of *T. turgidum* [19,57,58,60,61,62,63]. Unfortunately, none of these cited authors validated the species name of “*T. parvicoccum*”. Avoiding the question of a valid publication of “*T. parvicoccum*”, Kislev [57] proceeded in 2009 to list it as a subspecies of *T. turgidum* and clarified his original concept by including the more than 100,000 charred grains and approximately 100 rachis internodes found in a jar from Timnah (Tel Batash), Stratum VII (locus 437, storage jar 4120). This site is located on Israel’s inner coastal plain (Shfela), near the western foothills of the Judean Mountains (dated Late Bronze Age IIA, 14th century BCE). Of greater relevance, Kislev [57] (pp. 237–238) furnished differential characters that distinguish “*T. parvicoccum*” of the Near East and the Balkans, from the Central Asian species *T. compactum* Host and other hexaploid and tetraploid wheats.

### 3.2. Full Description of Triticum turgidum subsp. parvicoccum

According to the current systematics of the genus *Triticum* (see, e.g., [29,30,31]) and the cytogenetic data [25,26], we consider the subspecific rank as the most appropriate.

***Triticum turgidum* subsp. *parvicoccum*** Kislev, **subsp. nov.**—“*Triticum parvicoccum* Kislev” [nom. inval., *ICN* Art. 39.1]. “*Triticum turgidum* subsp. *parvicoccum* Kislev” [nom. inval., *ICN* Art. 39.1].—**Holotype**: Charred rachis fragment from Timnah (Tel Batash), on the inner Coastal Plain (Shfela) near the western piedmont of the Judean Mountains, Israel, deposited at the archaeobotanical Laboratory in Bar Ilan, National Natural History Collection.

For a complete description of this taxon, see Kislev [in *From Foragers to Farmers*: 237–238. 2009].

For an image of the holotype, see Figure 2A.

For images of the paratypes, see Figure 2B–E.

An image of the holotype was reproduced by Kislev ([42] in 1984 (p. 66 [“Figure 3a”]); [57] (p. 237 [“Figure 24.5”]) and Kislev et al. [58] ([“photo 142”]). Holotype and paratypes are deposited at the Bar-Ilan University, Archaeobotanical Laboratory, National Natural History Collection of Seeds and Fruits, established in the 1970s [64].

In claiming the species status of “*T. parvicoccum*”, Kislev in 1984 [42] summarizes: “Only recently were impressive ear and grain remnants of this wheat found at Tel Batash, Israel, dated to the 14th c. BCE, including glumes with a prominent keel along [their] length, characteristic of tetraploid wheats [65,66] (p. 37)”. This fragment is a charred rachis of *T. turgidum* subsp. *parvicoccum* that was described by Kislev in 2009 [57] (p. 237) as: “the culm upper part (on the bottom, ending in a collar) bears the lower part of the ear. The internodes are cuneate, the 5th is longer than the 3rd. The hairs on the rachis (along internodes sides and on the dorsal face of nodes) are long and well preserved. The 1st spikelet is very small and sterile, while the upper 4 are developed. Two prominent lumps beneath the glume bases are present at the 3rd, as well as at the 5th rachis nodes. Two glumes exhibit a developed keel along their length”.

The full description given below follows that provided in 2009 [57], considering that the information is given in a book that is not readily available. Between square brackets, items 2, 4, 6, and 9, we add the differences with the hexaploid *T. compactum* (cf. Percival in 1921: 307–320) [66].

“The grains were very short, oval to elliptic in shape, widest in the middle part of the grain or occasionally in the lower third. The apex was wide, rounded, or truncate and sometimes retuse. The germ was small and oval, the radicle prominent or slightly so, plumule slightly prominent or not at all and the cheeks were rounded. The ventral side was flat with a narrow to medium-wide crease (Figure 2B,D,E)”.

Additional characters (based on ear fragments found in the same archaeobotanical material):The culms are solid in the upper internode [not hollow as in hexaploid wheats], with a striate and slightly rough surface (Figure 2A). The first internode above the collar, which is almost round in cross section, bears an undeveloped spikelet and the second, short internode bears an infertile spikelet (Figure 2A);The rachis internodes are rather short, making the ear medium to high density [the ear is not very dense in hexaploid wheats]. The internodes, located on the lower part of the rachis, are shorter, thicker, and better preserved than others in the ear. The lower internodes are quite similar to the drawings of rachis remnants from Neolithic Tell Raman, SW Syria, published by van Zeist [67,68] and Zohary and Hopf [60] (p. 34) as tetraploid, free-threshing wheat;The ears are compressed laterally and are oblong in section; the 2-ranked side is wider than the face of the ear (Figure 2A and [58], photos 142–144);The rachis is fringed along the margins with long hairs [not short, as in hexaploid *T. compactum*]; a frontal tuft of long hairs is present on each node at the dorsal side of the spikelet base (Figure 2A);A pair of prominent lumps is present on the rachis node beneath its glume bases; after threshing, the basal part of the glumes usually remains attached to the rachis node [57], p. 34;The spikelets are narrow [not broad as in hexaploid wheats], two-flowered [not with three-five, grain-producing flowers];The glumes are hairy; their outer faces are somewhat flat;The glumes of the lateral spikelets are 7–13 mm long;The glume has a prominent keel running from the base to the tip [hexaploid have not a prominent keel except for the upper half] (Figure 2A and [58], photos 142–144, 148–149);

Kislev in 2009 [57] mentions other significant dissimilarities of various wheat remnants with small grains of “*T. parvicoccum*” from Timnah (Tel Batash):With the tetraploid *T. turgidum* subsp. *turanicum* (Jakubz.) Á. Löve and D. Löve [66] (p. 211). The ears are bearded with long awns, which were almost smooth near the base, and thereby similar to those of *T. turgidum* subsp. *durum* and different from those of *T. turgidum* subsp. *turanicum*;With the hexaploid *T. compactum* [66] (pp. 307–320). a. The geographical distribution of human-raised *T parvicoccum* is in the Near East and the Balkan, not Central Asia; and b. The angle between the glume and the rachis is less, not more than 45°, and the spikelets are not closely packed;

Details on Geographic Location—Tel Batash is located on an almost level plain in the wide alluvial Sorek Valley, close to the riverbed. As such, it differs from all other sites of the Shfela, which are located on hilltops. The valley is bounded on the north and south by low hills of the northwestern border of the Shfela. The region west of the site can be defined as the inner coastal plain, while to the east, the wide and fertile Sorek Valley stretches to the foot of the Judean Hills near Beth-Shemesh [69] (Figure 1).

## 4. Discussion

Kislev [38] considered that short round-grained samples of the subfossil wheat of Near East countries are not hexaploid but tetraploid with a genomic formula AABB. He later gave them the specific species name “*Triticum parvicoccum* Kislev” (see below), a species with a relatively compact ear and shorter grains. Feldman and Kislev in 2007 [62] mentioned that several remnants of this taxon (then indicated as “*T. turgidum* subsp. *parvicoccum*”) were found only in archaeological excavations in the Near East and the Balkans.

On the other hand, from an archaeobotanical point of view, “*T. parvicoccum* Kislev” is noteworthy as the most ancient of naked wheats. It is therefore the most likely species from which modern hexaploid bread wheat and possibly also tetraploid macaroni wheat descended [41,42,57]. The species was first published and described in 1980 [38]; see also [42,62,70,71]. Flourishing in early human agriculture, the taxon was grown in the early Neolithic (PPNB c. 8800-6500 BCE calibrated) across the Near East as far as the Balkans (c. 6500 BCE calibrated) [57].

Domesticated tetraploid wheats have significant genetic input from wild emmer populations in the southern Levant [72]. Wild tetraploid emmer wheat underwent parallel domestication, leading to the successive emergence of hulled domesticated emmer wheat (*T. dicoccum* Schrank ex Schübl. [≡ *T*. *turgidum* subsp. *dicoccum* (Schrank ex Schübl.) Thell.]) and large-seeded, free-threshing turgidum-durum wheat (*T. turgidum* sensu lato) [73]. Archaeological evidence suggests tetraploid wheats were first cultivated in this region before a pre-domesticated crop spread to southeast Turkey, where it interbred with northern Fertile Crescent wild emmer. Fixation of domestication traits in this mixed population explains observed allele sharing and genome-wide association study (GWAS) results. Feralization of pre-domesticated plants that lacked domestication traits likely produced modern wild populations in southeast Turkey, which combine traits of domesticates and southern Levant wild emmer. These populations appear to be the sole progenitor of domesticated tetraploids in treelike phylogenetic analyses [74].

According to Levy and Feldman [75], the primitive tetraploid *T. turgidum* subsp. *parvicoccum*, which was cultivated in the Levant around 9000 years ago and subsequently spread throughout the Fertile Crescent, is a strong candidate for contributing the BBAA genomes to *T. aestivum* [6,38,57]. This hypothesis is supported by Kerber’s [76] experimental work, where extracted tetraploids containing only the BBAA subgenomes from hexaploid wheat exhibited free-threshing characteristics and grain morphology similar to subsp. *parvicoccum* [38,57]. Although archaeological evidence shows subsp. *parvicoccum* persisted in regions beyond the Fertile Crescent, including the Balkans, Transcaucasia, and Georgia until approximately 1000 years ago [6], it became extinct during the Roman period, replaced by the more productive large-grained subsp. *durum* [57], although recently subsp. *parvicoccum* was reported from a medieval Abbasid site in Jerusalem [77]. Consequently, definitive identification of the naked, free-threshing domesticated tetraploid that hybridized with *Ae. tauschii* to produce *T. aestivum* remains elusive.

Archaeological plant-remains present unique challenges in taxonomic classification and interpretation within the broader context of botanical science. While these specimens are recovered from historical contexts and may be thousands of years old, they are not considered fossils in the traditional paleontological sense but rather are treated taxonomically as components of the extant flora, even in cases where the specific taxa are no longer living [9]. This distinction has important implications for both methodological approaches and theoretical frameworks in archaeobotanical research but also in clarifying the phylogeny of crops.

The process of identifying and classifying archaeobotanical specimens within modern taxonomic systems necessitates a multi-faceted approach incorporating several lines of evidence. Primary among these are detailed morphological analyses, which must consider both macroscopic and microscopic features [78]. Additionally, careful consideration of biogeographical context and established phylogenetic relationships provides crucial supporting evidence for taxonomic determinations. This integrated approach has become increasingly important as our understanding of plant domestication histories grows more nuanced [79,80].

Traditional assumptions about domestication traits and their expression in archaeological specimens have been challenged by recent findings. For instance, characteristics typically associated with domestication syndrome—such as non-shattering rachises in cereals or increased seed size—may not always follow expected patterns, necessitating careful interpretation of morphological data. This complexity is further illustrated by studies of ancient grain morphology in various archaeological contexts across Eurasia [81,82].

Recent research has demonstrated that the integration of multiple analytical approaches—combining traditional morphological studies with cutting-edge biomolecular techniques—provides the most robust framework for archaeobotanical classification. The field has been revolutionized by advances in molecular analytical techniques, particularly the ability to recover and analyze ancient DNA, proteins, and other biomolecules from archaeological plant-remains. These methodological developments offer unprecedented opportunities to establish taxonomic identities with greater precision and confidence. However, current technical limitations and preservation challenges necessitate a conservative approach to classification [83,84]. This cautious methodology is well-exemplified by the ongoing discourse surrounding the taxonomic status of archaeobotanical *T. aestivum* var. *antiquorum* (Heer) H. Messik. and its relationship to extant *T. sphaerococcum* subsp. *antiquorum* N.P. Gonch. [85,86].

For *Triticum turgidum* subsp. *parvicoccum*: The primary limitation lies in the carbonization state of the available specimens. While carbonized archaeobotanical remains can theoretically preserve ancient DNA (aDNA), several factors compromise its recovery and analysis: the carbonization process itself typically degrades DNA through heat-induced modifications; post-depositional processes over approximately thousands of years likely caused further DNA fragmentation; and potential contamination with modern DNA could complicate authenticity verification. Although specimens are available in the reference collection, the preservation state of biomolecules would likely be insufficient for reliable genomic analysis using current technologies.

For *T. aestivum* var. *antiquorum*: The situation is more fundamentally constrained by the complete absence of original material. Without physical specimens, molecular analysis is impossible regardless of technological capabilities. This absence creates a particularly challenging situation for taxonomic research, as the taxon’s classification must rely entirely on historical descriptions and morphological comparisons with similar materials from contemporary archaeological contexts.

In both cases, these limitations necessitate a classical taxonomic approach based on morphological characteristics, archaeological context, and historical documentation. Future advances in aDNA extraction and analysis techniques might eventually enable molecular studies of *T. turgidum* subsp. *parvicoccum* [87,88,89], but this remains currently unfeasible.

In this study, we considered the modern classification of tetraploid landraces and the intraspecific diversity of durum wheat (e.g., [90,91]). We also addressed the ongoing debate regarding the taxonomic treatment of *Triticum durum* and *Triticum turgidum*, specifically whether these should be classified as a single species or as two distinct species (e.g., [92,93,94,95,96,97]) adopting the first option. Based on the morphological evidence presented in Table 1 and the taxonomic context [53], treating *T. parvicoccum* as a subspecies of *T. turgidum* rather than an independent species is more appropriate for several compelling reasons. First, while *T. parvicoccum* shows some distinct features (smaller grain size, compact spike, and fragile rachis), these differences fall within the range of variation observed among other recognized subspecies of *T. turgidum*. The overall morphological pattern suggests modification of existing traits rather than development of novel characteristics that would warrant species-level distinction. Second, the archaeological context places it within the domestication trajectory of free-threshing tetraploid wheats, suggesting it represents a locally adapted form of *T. turgidum* rather than a separate evolutionary lineage. Finally, this taxonomic treatment better reflects its evolutionary relationship within the *T. turgidum* complex while still acknowledging its distinct morphological and archaeological significance through subspecific status.

Additionally, we examined the recent typification of *Triticum durum* through the selection of a North African durum wheat specimen from the Herbarium P (Desfontaines) as the lectotype for the name *T. durum* Desf. [98]. *T. turgidum* subsp. *parvicoccum* exhibits several distinctive morphological traits that clearly differentiate it from other free-threshing tetraploid wheats (Table 1). The taxon is characterized by its compact and short spike (>3.5 cm in length), which contrasts markedly with the longer spikes found in other subspecies (ranging from 4 to 16 cm). Its glumes are notably short and broad, differing from the elongated or intermediate shapes observed in other subspecies. The kernels are distinctively small (approximately 5 mm long) compared to the larger dimensions seen in other subspecies (ranging from 6 to 12 mm), and they exhibit an ovoid, ellipsoidal, or rounded shape. The spikelet characteristics are also distinctive, with a medium to high density, a length of 10–11 mm, and notably, only two florets per spikelet, which is fewer than all other subspecies (which range from two to seven florets). Additionally, it is the only subspecies described as having a fragile rachis, a trait that distinguishes it from the semi-fragile or non-fragile rachis found in other subspecies. These combined characteristics, along with Middle Eastern adaptation, create a unique morphological profile that clearly distinguishes *T. turgidum* subsp. *parvicoccum* from other tetraploid wheat subspecies.

**Table 1 genes-16-00274-t001:** Differential characteristics for the free-thresing tetraploid wheats under *Triticum turgidum* L. and the archaeobotanical taxa *T. turgidum* subsp. *parvicoccum* Kislev and *T. aestivum* (subsp. *compactum*) var. *antiquorum* (Heer) H. Messik.

Characters	*T. turgidum* subsp. *turgidum* (Rivet Wheat)	*T. turgidum* subsp. *durum* (Desf.) van Slageren (Durum Wheat)	*T. turgidum* subsp. *carthlicum* (Nevski) Á. Löve and D. Löve (Persian Wheat)	*T turgidum* subsp. *turanicum* (Jakubz.) Á. Löve and D. Löve (Khorassan Wheat)	*T. turgidum* subsp. *polonicum* (L.) Thell. (Polish Wheat)	*T. turgidum* subsp. *parvicoccum* Kislev (Small-Durum Wheat) *	*T. aestivum* (subsp. *compactum*) var. *antiquorum* (Heer) H. Messik. *
Spike shape	Dense, cylindrical	Dense, compact	Loose, elongated	Loose, elongated	Very long, lax	Compact, short	Dense, compact
Spike length (excluding awns)	7–11.5 cm	4–11 cm	9 cm	10–11.5 cm	(7) 10–16 cm	>3.5 cm	4 cm
Glume shape	Broad, rounded	Narrow, elongated	Intermediate	Intermediate	Long, narrow	Short, broad	Broad, rounded
Glume length	8–11 mm	8–12 mm	8–12 mm	12–15 mm	20–40 mm	7–13 mm	7–8 mm
Lemma awn	Awnless, short or long-awned, (7) 10–19 cm	Long-awned, (4) 20–23 cm	Short-awned, 8–11 cm	Medium-awned, 14–16 cm	Medium and prominent, 7–15 cm	Awnless	Awnless
Kernel color	Red or white	Amber (hard, vitreous)	Red or white	Red or white	Red or white	Unknown	Unknown
Kernel shape	Rounded, plump	Slender	Intermediate	Intermediate	Elongated	Ovoid, ellipsoidal or rounded	Bluntly rounded
Kernel dimensions	Intermediate, 6.7–8.4 mm	Long, 7–9.7 mm	Intermediate, 6–7 mm	Large, 10.5–12 mm	Large, 11–12 mm	Small, c. 5 mm	Small, 4–5 mm
Rachis hairs along the margins and frontal tuft at the base of each spikelet	White hairs, and a tuft of 1–2 mm-long white hairs	Long, and a tuft of hairs	Long, and a tuft of hairs	Few or no rachis hairs along the margins, generally lacks the frontal tuft of hairs	Hairs, and a tuft of hairs 2–2.5 mm long	Long, and a tuft of hairs 2 mm long	Unknown
Rachis fragility	Semi-fragile	Non-fragile	Semi-fragile	Semi-fragile	Semi-fragile	Fragile	ND
Spikelet density	High	High	Medium	Medium	Medium	Medium to high	High
Supernumerary spikelet spikes	Sporadic	Sporadic	Unknown	Sporadic	Unknown	Unknown	Unknown
Spikelet length	10–13 mm	10–15 mm	ND	15–17 mm	30–40 mm	10–11 mm	6–10 mm
Spikelet florets	4–7	5–7	2–4	2–4	4–5	2	3–5
Adaptation	Mediterranean and Temperate regions	Mediterranean regions	Caucasus, Middle East	Central Asia	Mediterranean, Central Asia	Middle East	Mediterranean and Temperate regions

Notes: * Only known in carbonized state. ND = No data available. References: [10,22,42,53,57,62,85,99,100,101,102].

The taxonomic classification of *Triticum parvicoccum* has been a subject of ongoing debate in the scientific literature. While some researchers recognize it as a distinct species, others classify it as a subspecies of *Triticum turgidum*. Additionally, some scholars reject any formal taxonomic status, instead considering it a short-grained variant of dense-eared forms of *T. durum* or *T. turgidum* [102,103]. However, based on the evidence presented in this study, we have adopted the subspecific rank for *T. parvicoccum*.

The reclassification of *Triticum parvicoccum* as *Triticum turgidum* subsp. *parvicoccum* offers profound insights into the history of wheat domestication and crop evolution, particularly given its status as an extinct archaeobotanical taxon known only from carbonized remains. This reclassification underscores the complexity of early agricultural systems, where farmers cultivated a wide range of plant species and varieties, many of which did not survive to the present day and highlights the diversity of pathways through which domestication may have occurred. As an extinct subspecies, *T. turgidum* subsp. *parvicoccum* represents a localized and ultimately unsuccessful experiment in wheat cultivation, yet its existence provides critical evidence of the experimentation and adaptation that characterized early farming communities in the Fertile Crescent.

The presence of *T. turgidum* subsp. *parvicoccum* in archaeological contexts suggests that early farmers were actively selecting and cultivating wheat varieties with desirable traits, such as non-shattering rachises and free-threshing characteristics. However, its extinction indicates that these traits alone were not sufficient to ensure its long-term survival or widespread adoption. *T. turgidum* subsp. *parvicoccum* may represent a localized domestication event that did not spread beyond its region of origin in West Asia. This highlights the importance of considering extinct taxa in models of crop evolution, as they provide evidence of early, unsuccessful domestication attempts [104]. This challenges the traditional view of domestication as a linear process, instead supporting a model of multiple, localized domestication events [105,106], some of which may have been evolutionary dead-ends. The reclassification of *T. parvicoccum* as a subspecies of *T. turgidum* aligns with genetic and morphological evidence, reinforcing the idea that free-threshing traits evolved independently in different populations of tetraploid wheat. This diversity of pathways reflects the adaptability of early agricultural systems and the role of human selection in shaping crop evolution.

The extinct status of *T. turgidum* subsp. *parvicoccum* also highlights the importance of archaeobotanical evidence in reconstructing the history of agriculture. Carbonized remains provide a snapshot of early farming practices, revealing the range of crops cultivated by ancient communities and the environmental conditions under which they thrived. By comparing *T. turgidum* subsp. *parvicoccum* with other extinct and extant wheat varieties, researchers can identify patterns of selection and adaptation that contributed to the domestication process. For example, the subspecies may have been adapted to specific ecological niches, such as arid or semi-arid regions, making it a valuable crop for early farmers in the Middle East. However, changes in climate, agricultural practices, or competition with other wheat varieties may have led to its eventual disappearance.

Several scenarios stay open for the role of *T. turgidum* subsp. *parvicoccum*:

As an early free-threshing wheat, the first scenario is the proto-domestication: *T. turgidum* subsp. *parvicoccum* may represent a proto-domesticated form of wheat, where early farmers selected for traits like non-shattering rachis and larger seeds but had not yet fully developed free-threshing characteristics. Its extinction suggests that these traits were not sufficient for long-term success. Another scenario is the local adaptation: This subspecies may have been adapted to specific environmental conditions, such as arid or semi-arid regions, making it a valuable crop for early agricultural communities in the Middle East. However, changes in climate or agricultural practices may have led to its extinction. A more stimulating scenario is its potential genetic contribution to modern wheats: Although *T. turgidum* subsp. *parvicoccum* is extinct, it may have contributed genetic diversity to other tetraploid and more likely hexaploid wheats through hybridization. But even this successful hybridization could be in the origin of its extinction [107]. This genetic reservoir could have provided traits for disease resistance, drought tolerance, or improved yield. Finally, a fourth scenario is that, as an extinct taxon, *T. turgidum* subsp. *parvicoccum* likely represents an evolutionary dead-end, a localized variety that did not contribute significantly to modern wheat but provides insights into the diversity of early agricultural systems.

This underscores the importance of studying extinct taxa as part of the broader narrative of crop evolution, as they provide insights into the genetic and phenotypic diversity that characterized early agricultural systems. Furthermore, the study of *T. turgidum* subsp. *parvicoccum* contributes to our understanding of domestication syndromes—the suite of traits that distinguish domesticated plants from their wild ancestors. These include non-shattering rachises, uniform seed size, and free-threshing characteristics, all of which were likely targets of early human selection.

The reclassification of *T. parvicoccum* also emphasizes the value of interdisciplinary approaches in crop science. By integrating genetic, morphological, agronomical, and archaeobotanical data, researchers can develop more nuanced models of crop evolution that account for the diversity of early agricultural systems. For example, while archaeobotanical evidence provides a temporal and spatial context for these changes, experimental cultivation of modern wheat varieties under conditions similar to those of early agriculture can further illuminate the agronomic traits of extinct taxa like *T. turgidum* subsp. *parvicoccum*.

Future research directions include expanding archaeobotanical surveys to identify additional sites where *T. turgidum* subsp. *parvicoccum* was cultivated, providing a clearer picture of its geographic and temporal distribution. Genome-wide analyses of carbonized remains can compare this subspecies with other *T. turgidum* varieties, shedding light on the genetic basis of free-threshing traits and other domestication-related characteristics. Comparative morphological studies can further clarify its place within the evolutionary history of tetraploid wheats, while experimental cultivation can recreate the conditions under which it may have been grown, offering insights into its agronomic potential.

In conclusion, the reclassification of *Triticum parvicoccum* as *Triticum turgidum* subsp. *parvicoccum* enriches our understanding of wheat domestication and crop evolution. As an extinct archaeobotanical taxon, it provides critical evidence of the diversity and complexity of early agricultural systems, highlighting the role of human selection and environmental adaptation in shaping crop evolution. By studying extinct varieties like *T. turgidum* subsp. *parvicoccum*, researchers can uncover valuable insights into the genetic and phenotypic diversity of early crops, informing modern breeding programs and contributing to the conservation of agricultural biodiversity. This reclassification not only deepens our understanding of the past but also underscores the importance of interdisciplinary approaches in addressing the challenges of contemporary agriculture.

## 5. Conclusions

The formal validation of *Triticum turgidum* subsp. *parvicoccum* Kislev addresses a critical taxonomic gap and yields three significant research applications. First, it provides a standardized framework for archaeobotanical classification, enabling more precise comparative analyses of ancient wheat assemblages. Second, the establishment of the Tel Batash holotype offers a definitive reference point for morphological studies of primitive free-threshing tetraploid wheat. Third, this taxonomic clarification strengthens our capacity to trace the evolutionary trajectory from early domesticated wheat to modern cultivars and might extend our understanding of wheat domestication and crop evolution.

Several urgent research priorities emerge from this validation. Comprehensive morphometric analyses of radiocarbon dated and georeferenced wheat remains across Southwest Asian sites are needed to map the temporal and spatial distribution of this subspecies. Additionally, comparative genomic studies between *T. turgidum* subsp. *parvicoccum* and contemporary wheat varieties could reveal previously unrecognized adaptive traits relevant to modern breeding programs. This research path may prove particularly valuable given increasing challenges from climate change and the need for resilient wheat varieties.

## Figures and Tables

**Figure 1 genes-16-00274-f001:**
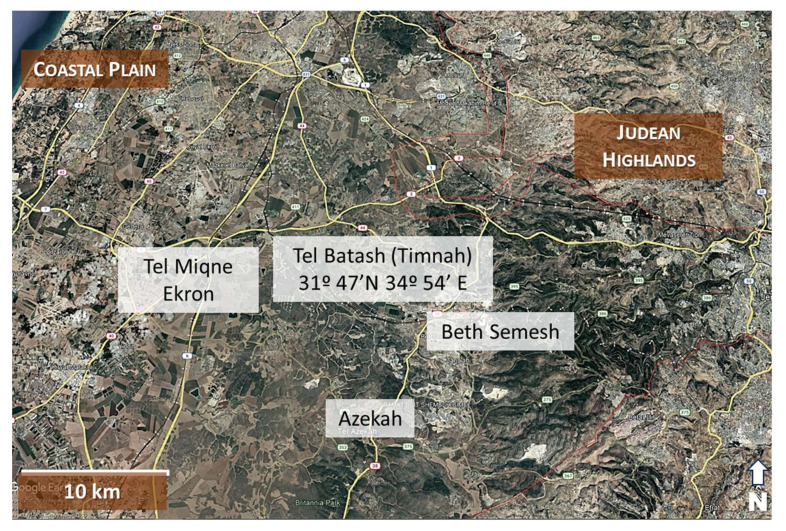
Regional situation map of Tell Batash (ancient Timnah) in the Shephelah region. The site (31°47’ N, 34°54’ E) occupies a strategic position between the coastal plain and Judean Highlands, controlling passage along the Sorek Valley.

**Figure 2 genes-16-00274-f002:**
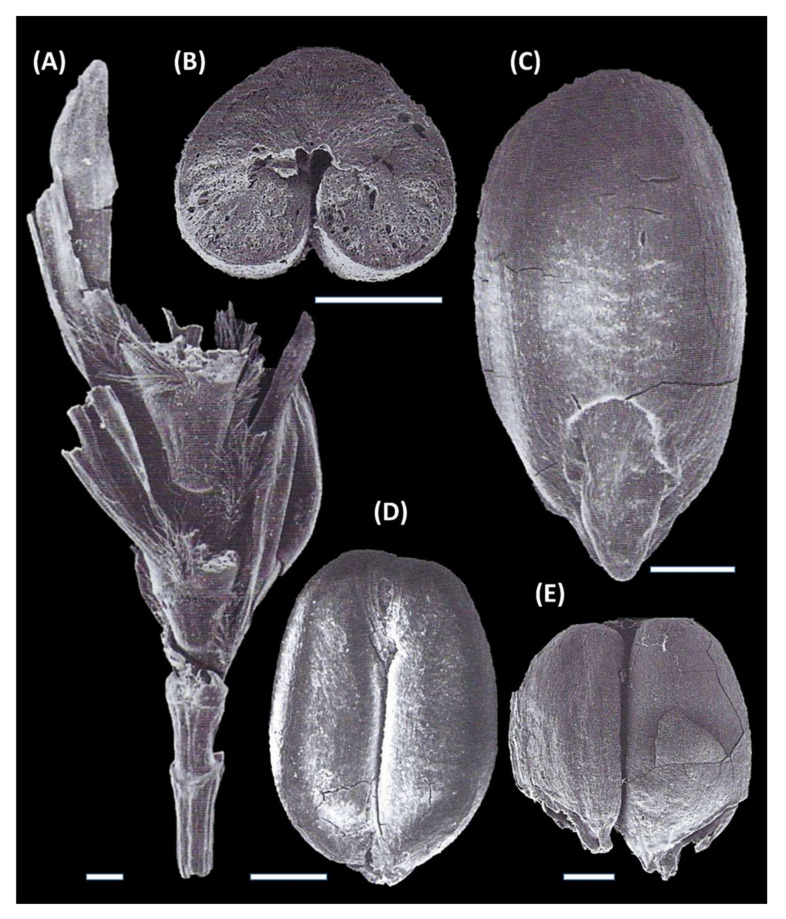
Image of the holotype and paratypes of *Triticum turgidum* subsp. *parvicoccum* Kislev. (**A**). Holotype, charred basal ear fragment, ventral view. It was found stored in a jar at Timnah (Tel Batash), Stratum VII, locus 437, basket 4.120. Image reproduced from Kislev in 1984 [42] (p. 66); in 2009 [57] (p. 237, Figure 24.5) and Kislev et al. in 2006 [58] (photo 142); (**B**). Image of the paratype, charred grain cross section; as for the other paratypes, it was found stored in the same jar, as above. Image reproduced from Kislev in 2009 [57] (p. 236), Figure 24.4 and Kislev et al. in 2006 [58] (photo 152); (**C**). Image of the paratype, charred grain dorsal view. Image reproduced from Kislev in 2009 [57] (p. 236, Figure 24.2) and Kislev et al. in 2006 [58] (photo 150); (**D**). Image of the paratype, charred grain ventral view. Image reproduced from Kislev in 2009 [57] (p. 236, Figure 24.3) and Kislev et al. in 2006 [58] (photo 151); (**E**). A pair of charred grains in their original position on a spikelet. Image reproduced from Kislev in 2009 [57] (p. 236, Figure 24.1). Specimens originate from a Late Bronze IIA storage vessel (Reg. No. 4120) [58], Pl. 41:9; containing wheat grains from Locus 437, Building 315, Stratum VII at Timnah (Tel Batash). Reference material is curated at the Archaeobotanical Reference Collection, Department of Land of Israel Studies and Archaeology, Bar-Ilan University. All SEM micrographs show carbonized specimens from this vessel. Scale bars: 1 mm.

## Data Availability

The original contributions presented in this study are included in the article. Further inquiries can be directed to the corresponding author.

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
