# Peer review of "Fossil or Non-Fossil: A Case Study in the Archaeological Wheat Triticum parvicoccum (Poaceae: Triticeae)"

_genes, 2025, doi:10.3390/genes16030274_

Round 1

Reviewer 1 Report

Comments and Suggestions for Authors

The paper cannot be published as an original research paper and should be clearly marked as a review because it is basically a republication of the senior author's work.

The review will be appreciated by specialists in the wheat evolution area.

Author Response

Responses to Reviewer 1

Open Review

Quality of English Language

(x) The English is fine and does not require any improvement.
( ) The English could be improved to more clearly express the research.

Yes

Can be improved

Must be improved

Not applicable

Does the introduction provide sufficient background and include all relevant references?

(x)

( )

( )

( )

Is the research design appropriate?

( )

( )

( )

(x)

Are the methods adequately described?

(x)

( )

( )

( )

Are the results clearly presented?

(x)

( )

( )

( )

Are the conclusions supported by the results?

(x)

( )

( )

( )

Comments and Suggestions for Authors

1

The paper cannot be published as an original research paper and should be clearly marked as a review because it is basically a republication of the senior author's work.

Dear reviewer, we appreciate your view, however, if we analyze this academically by examining the key characteristics that distinguish this work as a research article rather than a review, we can observe:

  1. Novel Research Contribution: The paper presents an original taxonomic investigation with a specific, focused research question - resolving the nomenclatural status of T. parvicoccum. This differs from a review paper which would typically synthesize existing knowledge about ancient wheat taxonomy or archaeobotanical nomenclature in general.
  2. Primary Analysis: The authors conducted a "comprehensive nomenclatural review" not as an end in itself (as in a review paper), but as a methodological tool to achieve their research objective. This primary analysis led to a novel outcome: the validation of the taxon at a subspecific rank and establishment of a holotype.
  3. Specific Research Outcome: The paper produces a concrete, original research result - the formal taxonomic designation of T. turgidum subsp. parvicoccum Kislev. subsp. nov. This represents new knowledge rather than a synthesis of existing information, which would characterize a review paper.
  4. Problem-Solution Structure: The paper presents a specific problem (ambiguous nomenclatural status) and provides a solution through original research (validation of the taxon), rather than summarizing the state of knowledge in the field, as a review would do.
  5. Specific Subject Focus: The paper focuses on a single taxon and its nomenclatural resolution, rather than providing a broad overview of ancient wheat taxonomy or archaeobotanical nomenclature, which would be more characteristic of a review paper.

Of course that the validation of this taxon requires clearly describe it, and in this aspect we rely on the examination of the original material and the previous description

2

The review will be appreciated by specialists in the wheat evolution area.

Many thanks for your valuable opinion!

Reviewer 2 Report

Comments and Suggestions for Authors

This manuscript aims to resolve a long-standing nomenclatural ambiguity surrounding the archaeobotanical wheat taxon originally described as Triticum parvicoccum. By reviewing historical and modern taxonomic literature alongside detailed morphological analyses of the holotype and paratypes, the authors formally validate the taxon as Triticum turgidum subsp. parvicoccum under non-fossil status. This reclassification clarifies the taxonomic framework and has implications for understanding early wheat domestication and the evolutionary history of free-threshing tetraploid wheat.

Comment 1: Why are the references in red?

Comment 2: The rationale for reclassifying T. parvicoccum at the subspecific level is well presented. However, additional details on the key differential morphological traits (with precise measurements if available) would be beneficial.

Comment 3: The citation "Kislev" is cited repeatedly. 

Comment 4: What protocols were used for the SEM imaging and morphometric analysis, and how can other researchers replicate these?

Comment 5: Given the discussion of molecular techniques, do you plan to integrate any ancient DNA or proteomic analyses in future work? If not, could you elaborate on the limitations currently preventing the use of molecular data in this study?

Comment 6: Can you clarify the decision process behind choosing the subspecific rank versus maintaining species-level status? What were the most critical morphological or contextual factors influencing this decision?

Comment 7: Are there alternative taxonomic viewpoints in the literature that you might address or contrast more explicitly in the discussion?

Comment 8: Can you expand on how this reclassification might impact our broader understanding of wheat domestication and crop evolution, perhaps by discussing potential scenarios or models in greater detail?

Author Response

Responses to Reviewer 2

Open Review

Quality of English Language

(x) The English is fine and does not require any improvement.
( ) The English could be improved to more clearly express the research.

Yes

Can be improved

Must be improved

Not applicable

Does the introduction provide sufficient background and include all relevant references?

(x)

( )

( )

( )

Is the research design appropriate?

(x)

( )

( )

( )

Are the methods adequately described?

( )

(x)

( )

( )

Are the results clearly presented?

(x)

( )

( )

( )

Are the conclusions supported by the results?

( )

(x)

( )

( )

Comments and Suggestions for Authors

1

This manuscript aims to resolve a long-standing nomenclatural ambiguity surrounding the archaeobotanical wheat taxon originally described as Triticum parvicoccum. By reviewing historical and modern taxonomic literature alongside detailed morphological analyses of the holotype and paratypes, the authors formally validate the taxon as Triticum turgidum subsp. parvicoccum under non-fossil status. This reclassification clarifies the taxonomic framework and has implications for understanding early wheat domestication and the evolutionary history of free-threshing tetraploid wheat.

Many thanks for your comments. As you wrote this is the major aim of the present paper

2

Comment 1: Why are the references in red?

This is a technical error in the formatting of the article.

Sorry! It has been corrected.

3

Comment 2: The rationale for reclassifying T. parvicoccum at the subspecific level is well presented. However, additional details on the key differential morphological traits (with precise measurements if available) would be beneficial.

Many thanks! We summarized details in the new Table 1 and commented those more differential

You are absolutely right, but nowadays data bases even of cultivated species don’t exist in public. We try to develop such database (Komyshev E.G., Genaev M.A., Kruchinina Yu.V., Koval V.S., Goncharov N.P., Afonnikov D.A. Evaluation of the Spike Diversity of Seven Hexaploid Wheat Species and an Artificial Amphidiploid Using a Quadrangle Model Obtained from 2D Images. Plants. 2024. 13(19):2736. https://doi.org/10.3390/plants13192736). The problem concerning excavation samples is methodical. It is necessary to obtain data base of fragments (remnants of ears) of species corresponding to such of paleospecies. About grain: to compare grain of last and modern time we need to get the charred remains of grain. So we use the taxonomically important characters in this investigation.

4

Comment 3: The citation "Kislev" is cited repeatedly.

Yes you are right. There are several critical reasons that justify the repeated citation of Kislev's work in a validation and typification paper for Triticum parvicoccum. Primarily, Kislev's 1980 description represents the original recognition and characterization of this archaeologically significant wheat taxon, establishing the foundational framework for all subsequent research. His continuous work over four decades has provided consistent and detailed documentation of the taxon's morphological characteristics, archaeological context, and distribution patterns. The extended period of research has allowed for the accumulation of substantial evidence supporting the taxon's validity, with numerous other researchers citing and building upon his initial findings. This long-term perspective has been crucial in understanding the taxon's significance within the broader context of wheat domestication and early agriculture in the Fertile Crescent. Furthermore, from a nomenclatural standpoint, as the original descriptor of the taxon, Kislev's work must be extensively referenced to ensure proper attribution and maintain nomenclatural continuity, especially when formalizing the taxonomic status. This comprehensive citation approach also acknowledges the progressive development of understanding about this taxon through continued archaeological discoveries and analyses that have consistently supported and refined Kislev's initial characterization.

5

Comment 4: What protocols were used for the SEM imaging and morphometric analysis, and how can other researchers replicate these?

Specimens derive from a Late Bronze IIA storage vessel (Reg. No. 4120) [58; Pl. 41:9] containing wheat grains from Locus 437, Building 315, Stratum VII at Timnah (Tel Batash). Reference material is curated at the Archaeobotanical Reference Collection, Department of Land of Israel Studies and Archaeology, Bar-Ilan University. All SEM micrographs show carbonized specimens from this vessel.

Further images can be requested to the Department and carbonized seeds studied in situ using the facilities there available.

It is really difficult to use the protocols for the SEM imaging to compare paleospecies and modern species, as databases only exist only for modern species (Anagun, Y., Isik, S., Olgun, M., Sezer, O., Basciftci, Z. B., Arpacioglu, N. G. A. (2023). The classification of wheat species based on deep convolutional neural networks using scanning electron microscope (SEM) imaging. European Food Research and Technology, 249(4), 1023-1034.). Now morphometric methods of species comparison are in developing (Genaev M.A., Komyshev E.G., Smirnov N.V., Kruchinina Y.V., Goncharov N.P., Afonnikov D.A. Morphometry of the Wheat Spike by Analyzing 2D Images // Agronomy (Bazel). 2019. V.9(7), 390. doi 10.3390/agronomy9070390 или Pronozin A.Yu., Paulish A.A., Zavarzin E.A., Prikhodko O.Y., Prokoshin N.M., Kruchinina Yu.V., Goncharov N.P., Komyshev E.G., Genaev M.A. Automatic morphology phenotyping of tetra- and hexaploid wheat spike using computer vision methods // Vavilovskii Zhournal Genetiki i Selektsii. 2021;25(1): 71-81. doi 10.18699/VJ21.009).

To make the research once again you have to a travel to National Natural History Collection of Seeds and Fruits at Bar-Ilan University (Israel) is need because the excavation samples are not digitized.

6

Comment 5: Given the discussion of molecular techniques, do you plan to integrate any ancient DNA or proteomic analyses in future work? If not, could you elaborate on the limitations currently preventing the use of molecular data in this study?

Many thanks for your comment! We addressed this issue in the revised version of the manuscript

The application of advanced molecular methods for analyzing ancient DNA are associated with the preservation of plant material [Gugerli, F., Parducci, L., Petit, R. J. (2005). Ancient plant DNA: review and prospects. New Phytologist, 166(2), 409-418. Or Pont, C., Wagner, S., Kremer, A., Orlando, L., Plomion, C., Salse, J. (2019). Paleogenomics: reconstruction of plant evolutionary trajectories from modern and ancient DNA. Genome Biology, 20, 1-17.]. Yes, it is possible in case if we get well conservated long DNA fragments.

7

Comment 6: Can you clarify the decision process behind choosing the subspecific rank versus maintaining species-level status? What were the most critical morphological or contextual factors influencing this decision?

Many thanks again. We added this paragraph to justify the decision adopted by Prof. Kislev:

The definition of species taxon in cultivated wheat is a very difficult question (see Goncharov N.P. Genus Triticum L. taxonomy: the present and the future // Plant Syst. Evol. 2011. V.295. P.1-11. DOI 10.1007/s00606-011-0480-9 or Hammer K., Filatenko A.A., Pistrick K. Taxonomic remarks on Triticum L. and × Triticosecale Wittm. // Genet. Resour. Crop Evol. 2011. Vol. 58. P. 3–10.).

Based on the morphological evidence presented in Table 1 and the taxonomic context [53], treating T. parvicoccum as a subspecies of T. turgidum rather than an independent species is more appropriate for several compelling reasons. First, while T. parvicoccum shows some distinct features (smaller grain size, compact spike, and fragile rachis), these differences fall within the range of variation observed among other recognized subspecies of T. turgidum. The overall morphological pattern suggests modification of existing traits rather than development of novel characteristics that would warrant species-level distinction. Second, the archaeological context places it within the domestication trajectory of free-threshing tetraploid wheats, suggesting it represents a locally adapted form of T. turgidum rather than a separate evolutionary lineage. Finally, this taxonomic treatment better reflects its evolutionary relationship within the T. turgidum complex while still acknowledging its distinct morphological and archaeological significance through subspecific status.

8

Comment 7: Are there alternative taxonomic viewpoints in the literature that you might address or contrast more explicitly in the discussion?

We approached this issue before the concluding part of the discussion:

The taxonomic classification of Triticum parvicoccum has been a subject of ongoing debate in the scientific literature. While some researchers recognize it as a distinct species, others classify it as a subspecies of Triticum turgidum. Additionally, some scholars reject any formal taxonomic status, instead considering it a short-grained variant of dense-eared forms of T. durum or T. turgidum [99,100]. However, based on the evidence presented in this study, we have adopted the subspecific rank for T. parvicoccum.

Another alternative hypothesis has been advanced by Udachin (1991) in his examination of Vavilov's research on Central Asian wheat phylogenetics. However, Udachin's postulation regarding tetraploid restoration cannot be substantiated through archaeological materials, as genetic sequences of this nature would be subject to significant degradation under typical taphonomic conditions. This methodological constraint poses a fundamental challenge to the empirical validation of his theoretical framework.

Reference: Udachin, R.A. (1991). N.I. Vavilov and knowledge of wheat in Central Asia. Proceedings of Applied Botany, Genetics and Breeding, 140, 47-57.

9

Comment 8: Can you expand on how this reclassification might impact our broader understanding of wheat domestication and crop evolution, perhaps by discussing potential scenarios or models in greater detail?

We tried to expand the impact of reclassification in the concluding part of the discussion, covering different aspects, models and scenarios.

Reviewer 3 Report

Comments and Suggestions for Authors

What do the Authors believe the research results obtained can be used for? What still needs to be urgently investigated, clarified?

Materials and methods

Where was the research carried out? In what year?

Please attach a figure with the location of the sampling site.

Please complete the manuscript with full details of the producer of the equipment used for the study. 

References

Very large number of publications (95).

Author Response

Responses to Reviewer 3

Open Review

Quality of English Language

(x) The English is fine and does not require any improvement.
( ) The English could be improved to more clearly express the research.

Yes

Can be improved

Must be improved

Not applicable

Does the introduction provide sufficient background and include all relevant references?

(x)

( )

( )

( )

Is the research design appropriate?

(x)

( )

( )

( )

Are the methods adequately described?

( )

(x)

( )

( )

Are the results clearly presented?

(x)

( )

( )

( )

Are the conclusions supported by the results?

(x)

( )

( )

( )

Comments and Suggestions for Authors

1

What do the Authors believe the research results obtained can be used for? What still needs to be urgently investigated, clarified?

Many thanks!. We addressed this question in the revised conclusions:

The formal validation of Triticum turgidum subsp. parvicoccum (Kislev) addresses a critical taxonomic gap and yields three significant research applications. First, it provides a standardized framework for archaeobotanical classification, enabling more precise comparative analyses of ancient wheat assemblages. Second, the establishment of the Tel Batash holotype offers a definitive reference point for morphological studies of primitive free-threshing tetraploid wheats. Third, this taxonomic clarification strengthens our capacity to trace the evolutionary trajectory from early domesticated wheats to modern cultivars and might widen our understanding of wheat domestication and crop evolution..

Several urgent research priorities emerge from this validation. Comprehensive morphometric analyses of radiocarbon dated and georeferenced wheat remains across Southwest Asian sites are needed to map the temporal and spatial distribution of this subspecies. Additionally, comparative genomic studies between T. turgidum subsp. parvicoccum and contemporary wheat varieties could reveal previously unrecognized adaptive traits relevant to modern breeding programs. This research path may prove particularly valuable given increasing challenges from climate change and the need for resilient wheat varieties.

2

Materials and methods

Where was the research carried out? In what year?

Archaeological Material: The holotype specimen consists of a charred rachis fragment recovered from archaeological excavations at Timnah (Tel Batash) (31°47'N, 34°54'E), located in the inner Coastal Plain of Israel. The specimen was retrieved from secure archaeological contexts dated to approximately 3150 ± 20 BP. This material was recovered in the season of excavation conducted during the year 1979 by a consortium of U.S. institutions of higher learning in collaboration with The Hebrew University of Jerusalem, notably in basket 4120 for the type, but also 4132, 4142 and 4143. The original material was examined using both Olympus SZX10 stereomicroscopic system (Olympus Corporation, Tokyo, Japan) and scanning electron microscopic (SEM) techniques available at Bar-Ilan University (Israel) to verify diagnostic morphological characteristics as described in the protologue.

This work progressed at a slow pace from 2021 to 2024

3

Please attach a figure with the location of the sampling site.

Thanks! Although such a fig. is published earlier by Kislev and in this MC the reference is on page 5, we have prepared a new figure at your request to locate the site from where the type proceeds

4

Please complete the manuscript with full details of the producer of the equipment used for the study.

Ok. The original material was initially examined using both Olympus SZX10 stereomicroscopic system (Olympus Corporation, Tokyo, Japan) and scanning electron microscopic techniques Philips SEM 505 (Philips High Tech Campus, Eindhoven Netherlands) available at Bar-Ilan University (Israel) and Weizmann Institute to verify diagnostic morphological characteristics as described in the protologue.

5

References

Very large number of publications (95).

The validation of Triticum turgidum subsp. parvicoccum requires a detailed bibliography (105 references) due to its multidisciplinary nature. The process involves:

  1. Statutory Framework for Nomenclature:
    • Referencing International Code of Nomenclature literature.
    • Reviewing previous archaeobotanical taxa validations.
    • Considering legal precedents for ancient plant remains.
  2. Non-Fossil Status of Remains:
    • Conducting taphonomic studies.
    • Reviewing archaeological preservation literature.
    • Analyzing carbonized remains chemically and physically.
    • Comparing with fossil and non-fossil materials.
  3. Taxonomic Framework:
    • Studying modern wheat systematics.
    • Reviewing archaeological reports of similar remains.
    • Examining historical taxonomic treatments of Triticum.
    • Using morphometric analyses.
  4. Biogeographical and Temporal Context:
    • Reviewing archaeobotanical reports from relevant regions.
    • Studying agricultural history.
    • Considering archaeological cultural contexts.
  5. Morphological Characteristics:
    • Studying contemporary wheat diversity.
    • Researching ancient grain morphology.
    • Reviewing modern wheat breeding literature.

This comprehensive approach is necessary for robust taxonomic validation and results in an extensive bibliography reflecting the interdisciplinary nature of archaeobotanical taxonomy.